# Potential Parasitic Causes of Epilepsy in an Onchocerciasis Endemic Area in the Ituri Province, Democratic Republic of Congo

**DOI:** 10.3390/pathogens10030359

**Published:** 2021-03-18

**Authors:** Melissa Krizia Vieri, Michel Mandro, Chiara Simona Cardellino, Pierantonio Orza, Niccolò Ronzoni, Joseph Nelson Siewe Fodjo, An Hotterbeekx, Robert Colebunders

**Affiliations:** 1Global Health Institute, University of Antwerp, 2610 Antwerp, Belgium; an.hotterbeekx@uantwerpen.be; 2Division Provincial de la Santé, P.O. Box 57, Ituri, Bunia, Democratic Republic of the Congo; Michel.MandroNdahura@student.uantwerpen.be; 3Department of Infectious Tropical Diseases and Microbiology (DITM), IRCCS Sacro Cuore Don Calabria Hospital, 37024 Negrar di Valpolicella (Verona), Italy; chiara.cardellino@sacrocuore.it (C.S.C.); pierantonio.orza@sacrocuore.it (P.O.); niccolo.ronzoni@sacrocuore.it (N.R.); 4Global Health Institute, University of Antwerp, Antwerp (Belgium) and Brain Research Africa Initiative (BRAIN), P.O. Box 25625, Yaoundé, Cameroon; JosephNelson.SieweFodjo@uantwerpen.be

**Keywords:** river epilepsy, onchocerciasis, *Onchocerca volvulus*, *Taenia solium*, *Strongyloides*, *Toxocara canis*, Democratic Republic of the Congo

## Abstract

A high burden of epilepsy is observed in Africa where parasitological infections are endemic. In 2016, in an Onchocerciasis endemic area in the Logo health zone, in Ituri province in the Democratic Republic of Congo, a door-to-door study showed an epilepsy prevalence of 4.6%, and 50.6% of persons with epilepsy were infected with *Onchocerca volvulus*. In the current study, the serum of 195 people infected with *O. volvulus* persons with epilepsy were tested to determine the proportion of co-infections with *Taenia solium, Toxocara canis* and *Strongyloides*. These proportions were, respectively, 8.2, 18.5 and 12.8%. Persons with a *T. solium* co-infection were older than those without co-infection (*p* = 0.021). In six (37.5%) of the *T. solium* co-infected persons, the first seizures appeared after the age of 30 years compared to three (2.1%) persons without a co-infection (*p* < 0.0001). Our study suggests that an *O. volvulus* infection is the main parasitic cause of epilepsy in the Ituri province, but in some persons, mainly in those with late onset epilepsy and with focal seizures, the epilepsy may be caused by neurocysticercosis. As the population in the area rears pigs, activities to limit *T. solium* transmission should be implemented.

## 1. Introduction

A higher burden of epilepsy is observed in low and middle incomes countries where parasitological infections are endemic. [1]. In sub-Saharan Africa, the two parasites that are well-established causes of epilepsy are *Plasmodium falciparum*, causing cerebral malaria [2], and *Taenia solium*, causing neurocysticercosis (NCC) [3]. Other parasites such as schistosomiasis, human African trypanosomiasis and toxocariasis are involved in a variety of neurological syndromes that may also present with seizures [1]. Furthermore, for a long time, it has been suggested that the *Onchocerca volvulus* parasite is associated with epilepsy [4,5]. However, it is only in recent years that epidemiological studies provided strong evidence for the association between onchocerciasis and epilepsy [6]. Therefore, the term onchocerciasis-associated epilepsy (OAE) was proposed [7,8]. This type of epilepsy often appears in previously healthy children between the ages of 3 to 18 years in onchocerciasis-endemic regions with high past or ongoing onchocerciasis transmission [9]. OAE presents with a wide clinical spectrum including generalized tonic–clonic seizures, nodding seizures, stunting and delayed development of secondary sexual characteristics [9,10]. To identify OAE, certain epidemiological and clinical criteria were proposed [11]: (1) the person has to live in an onchocerciasis endemic region for at least 3 years, (2) the onset of seizures has to occur between 3–18 years of age, (3) there is a high prevalence of epilepsy in the village and there are several families with more than one child with epilepsy in this village, (4) there is no obvious cause of epilepsy such as perinatal trauma, recent head trauma, cerebral malaria, encephalitis, (5) prior to the onset of epilepsy, the psychomotor development of the child was normal, and (6) the person presents onchocerciasis antibodies and/or microfilariae in skin snips. If a person meets the criteria above, but in addition presents with head nodding seizures (nodding syndrome) or Nakalanga features (stunted growth, eventually with thoracic or spinal abnormalities and/or with delayed sexual development) [12], even in the absence of an OV16 antibody or skin snip result, such a person should be considered as a person with OAE [9]. This OAE definition was shown to be useful in epidemiological studies to estimate the burden of epilepsy potentially caused by onchocerciasis and to identify hotspots where most likely onchocerciasis elimination programs are working sub-optimally [9]. However as long as there is no biomarker for OAE, meeting the criteria of this definition cannot exclude that the epilepsy is caused by another parasite such as *T. solium*.

So far, the pathophysiological mechanism of OAE remains unknown. Invasion of the *O. volvulus* parasite in the central nervous system is highly unlikely as no *O. volvulus* microfilariae nor DNA were detected in the cerebrospinal fluid of persons with OAE [13] nor in brain samples of persons who died with OAE [14]. Additionally, no *Wolbachia* DNA, an endosymbiont of *O. volvulus*, was detected in cerebrospinal fluid [14] and brain samples.

A high prevalence of epilepsy has been reported in many onchocerciasis areas but in many of the studies carried out in these areas, only persons with epilepsy were tested for the presence of either *T. solium* or *O. volvulus* infection [15]. In only a few studies, persons with epilepsy were tested for both parasitic infections [16].

In 2016, we documented a very high prevalence of epilepsy (4.6%) in onchocerciasis-endemic villages in the Logo health zone, in Ituri province in the Democratic Republic of the Congo (DRC) [17]. In this area, mass drug administration with ivermectin was never implemented. In 2015, in a case control study of 59 cases and 65 village controls of the same age groups, a strong association between epilepsy and *O. volvulus* infection was documented [18]. Indeed, 56% of cases compared to 26% of controls were *O. volvulus* skin snip positive. Moreover, the microfilariae load in skin snips was 3–10 times higher among cases than controls, and 51% of cases were OV16 antibody positive compared to 22% of controls. In 2017, we initiated a randomized trial in the same villages among persons with epilepsy to investigate whether ivermectin could decrease the frequency of their seizures [19]. During the screening phase of this trial, all persons with epilepsy who had signed an informed consent were skin snipped and tested for *O. volvulus* antibodies using the OV16 rapid test. Of the 399 persons with epilepsy tested, 202 (50.6%) presented either *O. volvulus* antibodies or a microfilarial positive skin snip or both. The latter were asked to participate in the trial. Blood samples of all trial participants were stored in a −80 °C deep freezer and later transferred to Europe respecting the cold chain.

To investigate the validity of the OAE definition and to estimate the proportion of persons infected with *O. volvulus* with epilepsy co-infected with another parasite, we retested all samples in Europe for the presence of *T. solium, Toxocara canis* antibodies. In addition, a *strongyloides* test was performed to investigate whether this infection is present in the Logo health zone, where a trial is planned to investigate the effect of moxidectin on *O. volvulus* infection and where potentially the effect of moxidectin on the *S. stercoralis* parasite could be investigated.

In this paper, we report the seroprevalence of these parasitic infections in persons infected with *O. volvulus* with epilepsy.

## 2. Materials and Methods

### 2.1. Study Design and Site

This was a cross-sectional study of persons with epilepsy in onchocerciasis-endemic villages in the Logo health zone, Ituri province, DRC

### 2.2. Study Population

In total 387 persons with epilepsy were screened for *O. volvulus* infection to select participants for a clinical trial to assess the effect of ivermectin on the frequency of seizures. In total, 195 persons infected with *O. volvulus* were tested for the presence of *T. solium, T. canis* and *Strongyloides*.

### 2.3. Diagnosis of O. volvulus Infection

*O. volvulus* antibodies were detected using the OV16 rapid diagnostic test (OV16 RDT, SD Bioline Onchocerciasis IgG4 rapid test, Abbott Standard Diagnostics, Inc., Yongin, Korea).

Skin snips were taken from the left and right iliac crests of participants using a sterile Holtz corneo-scleral punch (2 mm) to investigate infection with *O. volvulus*. The collected skin snips were incubated for 24 h in isotonic saline in a flat-bottomed microtiter plate. The microfilariae that emerged were counted using an inverted microscope, and the average count for both skin snips from each participant was calculated. Microfilarial densities were expressed as microfilaria/skin snip. The same experienced laboratory technician examined the skin snips from all study sites. We considered a person to have been infected with *O. volvulus* if the person either presented mf in skin snips and/or OV16 antibodies.

### 2.4. Testing for Other Parasites

Testing for *T. solium* antibodies was done with a commercial enzyme-linked immune electro transfer blot (EITB) test on serum (cysticercosis Western blot IgG) produced by LDBIO Diagnostics (Lyon, France). An ELISA test for *T. canis* antibodies was done with the DRG ^®^
*Toxocara canis* ELISA (EIA-3518) test (DRG International Inc., Springfield, NJ, USA). *Strongyloides* antibodies were tested using two tests: the CTD in-house immunofluorescence technique (IFAT) [20] and the commercial ELISA tests (Bordier ELISA) (Bordier Affinity Products SA, 1023 Crissier, Switzerland) [21]. As the former had proved to be highly sensitive but less specific in a diagnostic study, while the latter was on the contrary less sensitive but more specific, we relied on a combination of both tests in order to provide the final result [22]. All commercial tests were done according to the manufacturer’s guidelines.

Regarding *T. canis*, we interpreted the positivity or negativity of the test based on the ELISA results with a cutoff set at 1. Hence, all the quantification above or equal to 1 ng/mL were considered as positive. We considered a *Strongyloides* test positive if: (1) ELISA was positive (≥1) independently of the IFAT result; (2) IFAT was positive (any titer), and ELISA was higher or equal to 0.95; (3) ELISA was inferior to 0.95, and IFAT was higher or equal to 80. If none of these rules were fulfilled, we considered *Strongyloides* as negative.

### 2.5. Statistical Analysis

Data were analyzed using Microsoft Excel 2010 and IBM SPSS statistics version 27. Clinical and demographic variables were compared using the Fisher exact test (Chi-squared) for categorical variables and Mann–Whitney test for continuous variables. *p*-Values < 0.05 were considered significant.

### 2.6. Ethical Considerations

Ethical approval for the study was obtained from the ethic committee of the School of Public Health of the University of Kinshasa in the DRC (Approval number: ESP/CE/013/2018) and the ethical committee of the University of Antwerp (Registration number: B300201733350). All persons with epilepsy willingly participated in the study and provided signed/thumb-printed informed consents. The identity and information of participants was kept confidential.

## 3. Results

### 3.1. Description of the Population

Samples were examined from 195 persons infected with *O. volvulus* with epilepsy; mean age 25 years; 96 (50.8%) were men (Table 1). A higher proportion of persons infected with *O. volvulus* with epilepsy co-infected with *T. solium* were men compared to persons with *O. volvulus* infection only (Table 1). Persons co-infected with *T. solium* were also older than those with *O. volvulus* infection only. Focal seizures without loss of consciousness were more often reported in *T. solium* co-infected persons. No significant difference was observed concerning skin snip microfilarial positivity and microfilarial load (Table 1).

### 3.2. Seroprevalence of O. volvulus, T. solium, T. canis and S. stercoralis in Persons with Epilepsy

Of those 195, 16 (8.2%) presented *T. solium* antibodies using the EITB assay, 18.5% presented *T. canis* antibodies, and 12.8% presented *Strongyloides* antibodies (Table 2).

### 3.3. Characteristic of Co-Infected Persons with Epilepsy

Significantly more persons with epilepsy with only an *O. volvulus* infection were in the 10–20 years age group compared to *T. solium* co-infected persons, *p* = 0.011 (Figure 1). The age distribution of persons with epilepsy with only *O. volvulus* infection and persons co-infected with *T. canis* were similar.

The onset of seizures was between the ages 5–20 years in 125 (88%) of *O. volvulus* only infected persons compared to 8 (50%) of the *T. solium* co-infected persons, *p* < 0.0001 (Figure 2). In six (37.5%) of the *T. solium* co-infected persons, their first seizures appeared after the age of 30 (Table 3) compared to three (2.1%) persons with epilepsy without a co-infection, *p* < 0.0001 (Figure 2). The age of onset of seizures in persons with epilepsy with only an *O. volvulus* infection and in those with a *T. canis* co-infection was similar. Thirteen (81.3%) of *T. solium* co-infected persons with epilepsy were men in a total of 96 (50.8%) men in the entire population, *p* = 0.021.

Of the 195 *O. volvulus* infected persons with epilepsy, 48 (24.6%) were co-infected with either *T. solium* or *T. canis*. Of the 159 (81.5%) persons meeting the criteria of OAE, eight (5.0%) presented *T. solium* antibodies.

## 4. Discussion

In this study among persons infected with *O. volvulus* with epilepsy, we observed a relatively high proportion of individuals with serological evidence for a co-infection with another parasite: 8.2% for *T. solium*, 18.5% for *T. canis* and 12.8% for *S. stercoralis*. However, this does not mean these other parasites played a causative role in the epilepsy. Indeed, *S. stercoralis* is not known to be a cause of epilepsy.

*T. solium* clearly is a potential cause of epilepsy in the Logo health zone. However, *T. solium* does not seem the main cause of epilepsy in this area. Only 8.2% of the persons infected with *O. volvulus* presented *T. solium* antibodies, while the *O. volvulus* seroprevalence among persons with epilepsy in the area was 50.6%. It may be that the *T. solium* prevalence among persons with epilepsy without *O. volvulus* infection was higher than 8.2%. However, only 5% of the persons with epilepsy meeting the OAE criteria presented *T. solium* antibodies. This shows the value of the OAE definition in the study area to identify persons with epilepsy potentially triggered by an *O. volvulus* infection and not by another parasitic infection.

Thirteen (81.3%) persons co-infected with *T. solium* with epilepsy were men, while only 96 (50.8%) of all persons with epilepsy in the study were men. This is in contrast with NCC, which appears to be more common among women [23]. The onset of seizures in *T. solium* co-infected persons more often occurred at a later age compared to *O. volvulus* infected persons without co-infection. This confirms that NCC is characterized by a later onset of epilepsy with more persons experiencing their first seizure after the age of 20 years [24].

It is unclear in how many of the 16 persons co-infected with *T. solium* NCC was the cause of the epilepsy. Most likely this was the case for the six persons with epilepsy in which the seizures started after the age of 20 years but was unlikely to be the cause of epilepsy in the person who developed nodding seizures at the age of eight. We observed a higher percentage of focal seizures (12.5%) in persons co-infected with *T. solium*. Focal seizures are often reported in persons with NCC because of one or more cysticercus brain lesions [25].

We used the EITB *T. solium* antibody test, a test with 98% sensitivity and 100% specificity for the diagnostic of cysticercosis [26]. However, without brain imaging studies, we are unable to determine the proportion of EITB *T. solium* positive individuals who actually have NCC [27]. Immuno-diagnosis of NCC is complex and strongly influenced by the course of infection, the disease burden, the cyst location and the immune response of the host [28]. A positive EITB result can be found in up to 20 to 25% of some rural populations, where the parasite is endemic [29]. However, as described in endemic regions of Peru and Columbia, a person seropositive with *T. solium* can become seronegative one year later (about 40% seropositive persons became seronegative), meaning that the group of people with only transient antibodies may have been exposed to *T. solium* but did not develop a viable infection, or they may have had cysticercosis [30]. In addition, a limitation of the EITB is its low sensitivity in patients with a single intracranial cysticercus and in those with only calcified parasites, where up to 50–70% of cases may be falsely negative [26,28]. Consequently, results of the EITB must be evaluated with caution. Currently, the real sensitivity and specificity of EITB for the diagnostic of NCC, in an area where *T. solium* is highly prevalent, is unknown.

A small survey in the study area in 2021 revealed that six (17.6%) of 36 households reared pigs. Moreover, these pigs are able to run free in the village, and there is a lack of latrines. Therefore, with a *T. solium* prevalence among persons with epilepsy of at least 8.2%, further assessment of the neurocysticercosis prevalence in the area should be considered.

*T. canis* does not seem to play a causative role in triggering epilepsy in our study population. The age profile of persons with epilepsy and the age at onset of seizures in *T. canis* co-infected persons with epilepsy, in contrast with persons co-infected with *T. solium*, was similar with persons with only an *O. volvulus* infection. This suggests that either the *T. canis* infection was not a causal factor of the epilepsy or that it is causing epilepsy in similar age groups, as *O. volvulus*. Toxocara larvae are able to cross the blood–brain barrier and invade the central nervous system, and serological studies suggest a potential association between Toxocara infections and epilepsy [31]. However, it is considered that *T. canis* is not a frequent cause of epilepsy [32].

Multiple parasitic infections were shown to increase the risk for acute convulsive epilepsy in sub-Saharan Africa [33]. In our study, only 24.6% of persons with epilepsy presented serological evidence of more than one parasitic infections.

It is important to mention the limitations of our study. The main limitation is that we did not include persons with epilepsy without *O. volvulus* infection and healthy controls. In addition, there is a possibility that cross-reactivity of the serological test for *Onchocerca*, *Strongyloides* and *Toxocara* may have influenced the results. However, correlation analysis did not show a positive significant correlation between the tests. Therefore, we do not know the prevalence of *T. solium, T. canis* and *Strongyloides* infection in all persons with epilepsy in the area and in the general population. Moreover, imaging studies were not performed and are needed to confirm a diagnosis of NCC.

## 5. Conclusions

In conclusion, our study suggest that an *O. volvulus* infection is the main parasitical cause of epilepsy in the onchocerciasis-endemic villages in the Logo health zone in Ituri, but in some persons, mainly in those with late onset epilepsy (after the age of 20 years) and in persons with focal seizures, the epilepsy may be caused by NCC. As the population in the area rears pigs, further assessment of the neurocysticercosis prevalence in the area should be considered.

## Figures and Tables

**Figure 1 pathogens-10-00359-f001:**
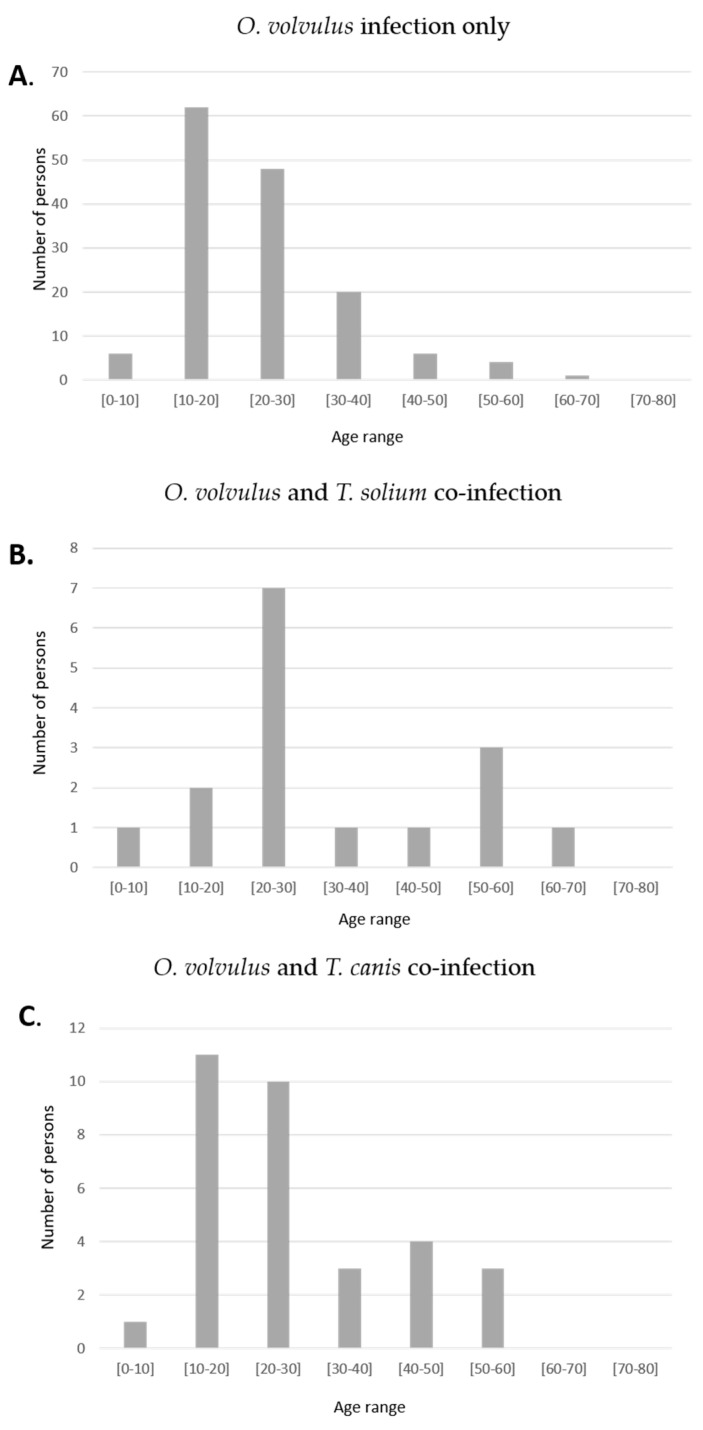
Age distribution of persons with *O. volvulus* infection only (**A**) with an *O. volvulus* and *T. solium* co-infection (**B**) and with an *O. volvulus* and *T. canis* co-infection (**C**).

**Figure 2 pathogens-10-00359-f002:**
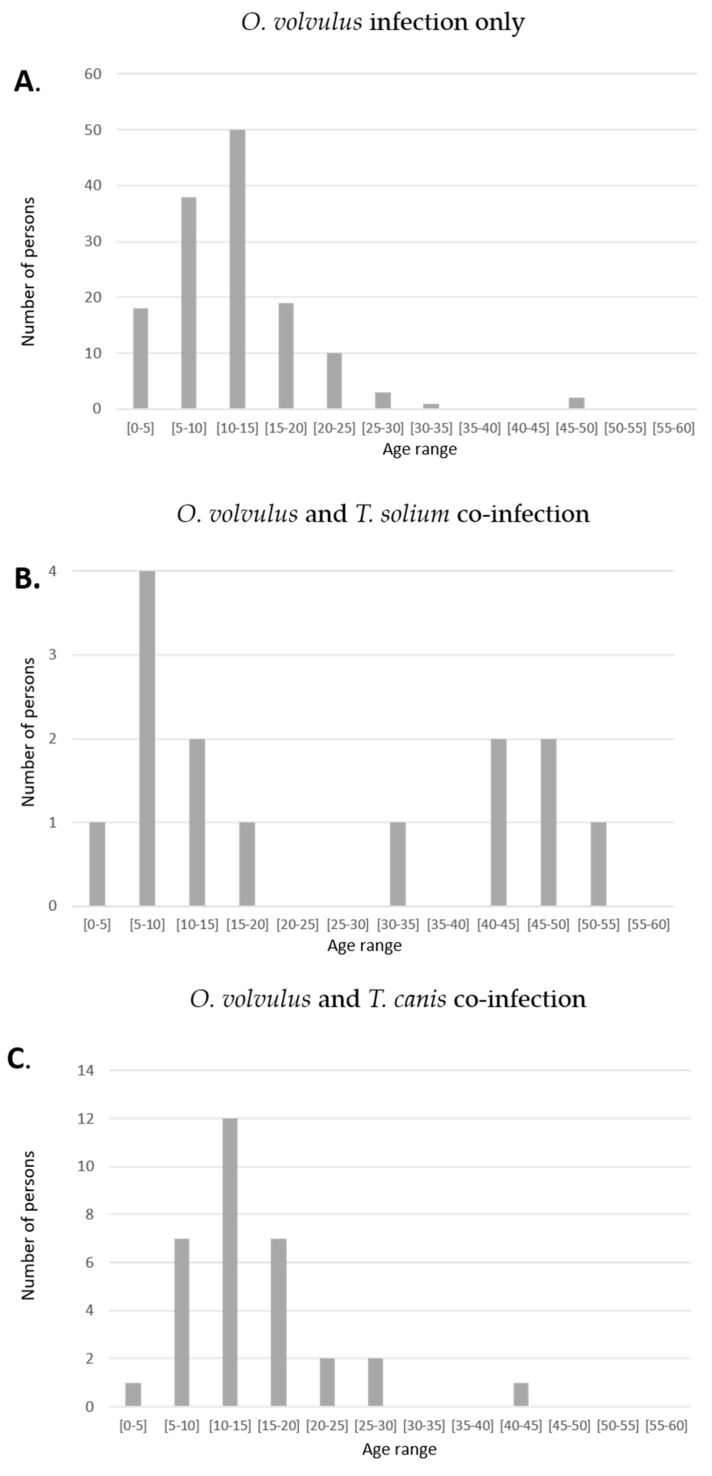
Age of seizure onset of persons with *O. volvulus* infection only (**A**) with an *O. volvulus* and *T. solium* co-infection (**B**) and with an *O. volvulus* and *T. canis* co-infection (**C**).

**Table 1 pathogens-10-00359-t001:** Characteristics of 195 persons with epilepsy with *O. volvulus* co-infected with *T. solium,* co-infected with *T. canis* and without co-infection.

	*O. volvulus* and *T. solium* Infection (N = 16)	*O. volvulus* and *T. canis* Infection (N = 32) ^2^	*O. volvulus* Only ^1^ (N = 147)	*p*-Value *	*p*-Value **
Age, Median (IQR)	27 (21–51)	23.5 (16-35)	22 (16–30)	0.021	0.130
Men, N (%) ^3^	13 (81.3%)	19 (59.4%)	71 (48.3%)	0.012	0.26
Age at onset of the epilepsy, Median (IQR)	14 (9–45)	13.5 (10–19.5)	12 (8–15)	0.122	0.051
Generalised tonic–clonic seizures, N (%)	14 (87.5%)	32 (100%)	131 (89.1%)	0.845	NA
Nodding seizures, N (%)	1 (6%)	2 (6.3%)	24 (16.3%)	0.288	0.143
Absences, N (%)	7 (43.7%)	9 (28.2%)	63 (42.9%)	0.945	0.123
Focal seizures with loss of consciousness, N (%)	2 (12.5%)	0(0%)	25 (17%)	0.523	<0.0001
Focal seizures without loss of consciousness, N (%)	2 (12.5%)	1 (3.1%)	3 (2%)	0.021	0.707
Skin snip, N (%)	13 (81.3%)	28 (87.5%)	115 (78.23%)	0.78	0.236
Mf load, Median (IQR)	4 (1.25–11.25)	10 (1–10)	9 (0–67)	0.385	0.907

IQR = interquartile range, N= number of persons, Mf = microfilarial, ^1^ strongyloide-positive persons were included because Strongyloides was not considered to be a cause of epilepsy. ^2^ Without *T. solium* infection. ^3^ 6 missing values. * *p*-value *O. volvulus versus T. solium*. ** *p*-Value *O. volvulus* versus *T. canis*.

**Table 2 pathogens-10-00359-t002:** Seroprevalence of *T. Solium*, *T. canis* and *Strongyloides* in persons with epilepsy with *O. volvulus* infection.

	Seroprevalence (N = 195)
***T. solium*** (WB), N (%)	16 (8.2%)
***T. canis*** (ELISA), N (%)	36 (18.5%)
***Strongyloides*** (IFAT + ELISA), N (%)	25 (12.8%)

**Table 3 pathogens-10-00359-t003:** Characteristics of persons with epilepsy infected with a *T. solium* and *O. volvulus* co-infection.

Nr	Age	Sex	Age of Seizures Onset	Generlized TCS	Nodding Seizures	Absences	Focal Seizures
1	10	M	8	YES	YES	YES	YES *
2	19	F	5	YES	NO	YES	NO
3	20	M	20	YES	NO	NO	YES **
4	21	M	14	YES	NO	YES	NO
5	22	F	9	YES	NO	NO	NO
6	23	M	9	YES	NO	NO	YES ***
7	24	M	7	YES	NO	NO	NO
8	27	M	11	YES	NO	YES	NO
9	27	M	12	YES	NO	YES	NO
10	29	M	10	YES	NO	YES	YES *
11	34	M	31	YES	NO	NO	NO
12	50	F	44	YES	NO	YES	NO
13	51	M	49	YES	NO	NO	NO
14	52	M	45	YES	NO	NO	NO
15	60	M	53	YES	NO	NO	NO
16	65	M	47	YES	NO	NO	NO

M = male, F = female, TCS = tonic–clonic seizures, NA = not applicable, * focal motor seizures without loss of consciousness, ** focal seizures with secondary bilaterally tonic–clonic seizures, *** focal motor seizures with loss of consciousness.

## Data Availability

The datasets generated during the current study are available from the corresponding authors on reasonable request.

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
