# Peer review of "Potential Parasitic Causes of Epilepsy in an Onchocerciasis Endemic Area in the Ituri Province, Democratic Republic of Congo"

_pathogens, 2021, doi:10.3390/pathogens10030359_

Round 1

Reviewer 1 Report

This again a professionally written, well documented article reporting results of a very interesting research.

For details, see attached file. 

Author Response

For details, see attached file

Reviewer 2 Report

The article by Vieri and colleagues provides well-written and logically structured important information about the possibility of neurocysticercosis (NCC) as cause of epilepsy in an region were onchocerciasis-associated epilepsy (OAE) has a high prevalence. Linking age groups and age of epilepsy onset supports the possibility of NCC playing a causative role, especially in later onset epilepsy rather than OAE being the cause.

I have some remarks:

In the discussion (line 209) it is explained that why no strongyloides was not included in table 1, fig. 1 and fig. 2. This explanation should come earlier since readers might question during the whole article why strongyloides was ignored in these tables and graphs.

Line 254: The age profile of persons with epilepsy and the age at onset of seizures in T. canis co-infected persons with epilepsy, in contrast with T. solium co-infected persons, was similar with persons with only an O. volvulus infection. This suggests that either the T. canis infection was not a causal factor of the epilepsy or that it is causing epilepsy in similar age groups as O. volvulus. Toxocara larvae are able to cross the blood-brain barrier and invade the central nervous system and serological studies suggest a potential association between Toxocara infections and epilepsy [24]. However it is considered that T. canis is not a frequent cause of epilepsy

Above reasoning would however be an argument to include strongyloides in the calculations cause the hypothesis it would give similar results as Toxocara (Strongyloides can also be found in the brain but is not a likely cause of epilepsy). Similar or non-similar results would confirm or reject this hypothesis.

Line 46 : ‘nodding seizures (nodding syndrome) or Nakalanga features (stunted growth, eventually with thoracic or spinal abnormalities and/or with delayed sexual development)[8], even in the absence of an OV16 antibody or skin snip result such a person should be considered as a person with OAE A brief explanation why this should be considered would be useful

Table 2 mentions 36 T. positive sample, whereas table 1 mentions 32: from where comes this difference?

The author honestly mention one of the limitations of their study: the lack of a control group. Also the doubt about sensitivity and specificity of serology are mentioned as a limitation. What is not mentioned as a limitation is the possibility of cross reactivity between different nematodes: Onchocerca is a filaria and filaria are roundworms or nematodes. So possible cross reactions of onchocerca with strongyloides or toxocara and cross reaction between strongyloides could have occurred with 1 of the 2 serologies false positive. Where there patients that had onchocerca + strongyloides + toxocara positivity and might thus indicate strongyloides/toxocara cross reactions?

It would have been interesting to not only have a non-epileptic control group, but to also test the 197 (399 – 202) patients of the 399 with epilepsy that where not diagnosed with OAC. This could shed light on cross-reactivity and possible causes op epilepsy. But I understand this would be a lot of work and might be done in a future study.

Reviewer 3 Report

This article shows that in the democratic republic of Congo,  in a population with onchocerciasis, epilepsy was not associated with toxocariasis and cysticercosis, the main parasitic causes of epilepsy. The results are interesting since it confirms the results of other studies from others countries. The paper is well written and the main limits of the study well described. However, some papers important in the field such as (Pion SD, Kaiser C, Boutros-Toni F, Cournil A, Taylor MM, Meredith SE, Stufe A, Bertocchi I, Kipp W, Preux PM, Boussinesq M. Epilepsy in onchocerciasis endemic areas: systematic review and meta-analysis of population-based surveys. PLoS Negl Trop Dis. 2009 Jun 16;3(6):e461. doi: 10.1371/journal.pntd.0000461. PMID: 19529767; PMCID: PMC2691484.) are not mentioned.

Author Response

We apologized that we have not introduced  these important papers in our  manuscript. We now included these references.